# Multiplex Immunofluorescence Assay of Infiltrating Mononu-Clear Cell Subsets in Acute T-Cell-Mediated Rejection and BK Virus-Associated Nephropathy in the Allograft Kidney

**DOI:** 10.3390/diagnostics12020268

**Published:** 2022-01-21

**Authors:** Mee-Seon Kim, Jeong-Hoon Lim, Man-Hoon Han, Sang-Yeob Kim, Yun Jae Kim, Yong-Jin Kim

**Affiliations:** 1Department of Pathology, School of Medicine, Kyungpook National University, Kyungpook National University Hospital, Daegu 41944, Korea; kimm2342@gmail.com (M.-S.K.); one-many@hanmail.net (M.-H.H.); 2Division of Nephrology, Department of Internal Medicine, School of Medicine, Kyungpook National University, Kyungpook National University Hospital, Daegu 41944, Korea; jh-lim@knu.ac.kr; 3Department of Convergence Medicine, University of Ulsan College of Medicine and Asan Medical Center, Seoul 05505, Korea; sykim3yk@amc.seoul.kr; 4Asan Institute for Life Sciences, Asan Medical Center, Seoul 05505, Korea; windkim87@gmail.com

**Keywords:** multiplex immunofluorescence assay, image analysis, T-cell-mediated rejection, BK virus-associated nephropathy, transplant kidney

## Abstract

Renal allograft biopsy is the gold standard procedure for diagnosis of kidney rejection via specific pathological changes. To provide a better assessment of immunologic events in acute T-cell-mediated rejection (acute TCMR) and BK virus-associated nephropathy (BKVAN) cases, we used multiplex immunofluorescence staining to identify infiltrating mononuclear cell subsets in the cortex area of transplanted kidneys. Antibodies to CD4, CD8, CD20, CD68, Foxp3, and cytokeratin were used. In cortical interstitium, CD8+ cells were significantly more prevalent in acute TCMR than BKVAN cases (34% vs. 22.8%, *p* = 0.034). In medulla, CD20+ cells were significantly more prevalent in BKVAN than acute TCMR cases (51.9% vs. 11.3%, *p* = 0.028).

## 1. Introduction

Renal allograft biopsy is the gold standard procedure for diagnosis of kidney rejection via specific pathological changes. The major factor in acute T-cell-mediated rejection (TCMR) is tissue infiltration of mononuclear cells. Acute TCMR severity depends on the percentage of the interstitial area affected by infiltration [1,2,3]. T cells are known to be important in the pathogenesis of acute TCMR, although multiple other mononuclear cell subpopulations, including B cells, NK cells, plasma cells, and monocytes/macrophages, have been shown to contribute to the outcome of rejection-related immunologic events [4,5,6,7]. However, cell population analysis is complicated using traditional immunohistochemistry because of the limitations of staining using multiple antibodies and human error in cell counting [8,9].

To provide a better assessment of immunologic events in acute TCMR and BK virus-associated nephropathy (BKVAN) cases, we used multiplex immunofluorescence assays to identify infiltrating mononuclear cell subsets in a transplanted kidney in cortex areas. Because BKVAN has characteristic interstitial mononuclear cell infiltration, and even tubulitis, which is one of the symbolic changes of acute TCMR, it is difficult to differentiate between the two conditions. Additionally, we tried to evaluate the differences in infiltrating cell subsets in the excluded areas for diagnosis, including fibrotic areas, the immediate subcapsular cortex, and adventitia around large vessels and medullary areas [1]. 

## 2. Materials and Methods

### 2.1. Study Design

Multiplex immunofluorescence assays of formalin-fixed, paraffin-embedded tissues (FFPE) were conducted using samples from nine patients with acute TCMR and five patients with BKVAN at various time points after transplantation. All samples were collected between 1 January 2015 and 1 December 2017 at the Department of Pathology in Kyungpook National University Hospital and Yeungnam University Medical Center in Daegu, Korea. The study protocol was approved by the Daegu Joint Institutional Review Board (DGIRB 2017-08-001), and informed consent was waived by the Daegu Joint Institutional Review Board. All methods were carried out in accordance with the Korean Bioethics and Safety Act. 

### 2.2. Patient Selection Criteria and Tissue Pathology Grading 

Inclusion criteria for acute TCMR (*n* = 9) were: category 4 histologic features of acute TCMR based on the Banff 2017 classification system [10]; negativity for SV-40 cells based on IHC; absence of BK virus in serum or urine; negativity for the donor-specific antibody. The male-to-female ratio of patients with TCMR was 6:3. The mean patient age and graft age were 46.8 (range, 33–62) years and 7.2 (1–15) months, respectively. Acute TCMR grades 1A, 1B, and 2B were observed in three different patients. Plasma cell-rich types of TCMR were not included (Table 1). Inclusion criteria for BKVAN (*n* = 5) were: histologic features of BKVAN with SV-40-positive cells in the cortex; clinical improvement in graft function with anti-viral treatment after pathologic diagnosis; no histologic features of acute TCMR or ABMR; >500 inflammatory cells in the cortical area. All BKVAN patients were male (*n* = 5). The mean patient and graft ages were 47.4 (25–62) years and 5.1 (2.5–7) months, respectively. According to the Banff Working Group Classification System [11], one patient had class A, three had class B, and one had class C diseases (Table 2).

### 2.3. Multiplex Immunofluorescence (IF) Assay

The formalin-fixed, paraffin-embedded (FFPE) blocks of renal biopsy specimens were cut into 2 μm sections. The slides were heated for at least 1 h in a dry oven at 60 °C and dewaxed using Leica Bond Dewax (#AR9222, Leica Biosystems). Then, multiplex immunofluorescence assays were performed using Leica Bond Rx Automated Stainer (Leica Biosystems, Newcastle, UK). Antigens were retrieved using Bond Epitope Retrieval 2 (#AR9640, Leica Biosystems) in a solution at pH 9.0 for 30 min. The slides were incubated with primary antibodies for CD8 (MCA1817T, dilution 1:300; Bio-Rad Laboratories, Hercules, CA, USA) for 30 min and then analyzed using polymer HRP Ms+Rb (ARH1001EA; PerkinElmer, Waltham, MA, USA) for 10 min. CD8 was visualized using Opal 690 TSA Plus (dilution 1:150; 10 min). To remove bound antibodies before the next step in the staining sequence, slides were treated with Bond Epitope Retrieval 1 (#AR9961, Leica Biosystems) for 20 min. In a serial fashion, the slides were incubated with the next primary antibodies against Foxp3 (ab20034, dilution 1:100; Abcam, Cambridge, UK) for 30 min and analyzed using Polymer HRP Ms+Rb. Foxp3 was also visualized using Opal 650 TSA Plus (dilution 1:300) for 10 min. The same procedure was repeated for staining with anti-CD20 (ab9475, dilution 1:50; Abcam; visualized with Opal 620 TSA Plus (dilution 1:150)), anti-CD68 (M0876, dilution 1:100; Dako, Santa Clara, CA; visualized with Opal 570 TSA Plus (dilution 1:150)), anti-CD4 (ab133616, dilution 1:100; Abcam; visualized with Opal 540 TSA Plus (dilution 1:300)), and anti-CK (M3515; dilution 1:500; Dako; visualized with Opal 520 TSA Plus (dilution 1:150)) immunoreagents. After treatment with Bond Epitope Retrieval 1 for 20 min, cell nuclei were subsequently visualized with 4′,6-diamidino-2-phenylindole (DAPI) stain, and the section was cover-slipped with HIGHDEF^®^ IHC Fluoromount (ADI-950-260-0025; Enzo Life Science, Inc., Farmingdale, NY, USA).

### 2.4. Image Acquisition and Quantitative Data Analysis

The slides were scanned using the PerkinElmer Vectra 3.0 Automated Quantitative Pathology Imaging System at 20 nm wavelength intervals from 420 to 720 nm. The different depth images were combined to create a single stack image, which retained the unique spectral signature of all multiplex immunofluorescence markers. The final image files were created using Vectra and analyzed using InForm 2.2.1 and TIBCO Spotfire software (PerkinElmer). To compare reliable unmixed images, the representative images of each emission spectrum and unstained tissue slides were used. Each individually stained section (CD8-Opal 690, Foxp3-Opal 650, CD68-Opal 620, CD20-Opal 570, CD4-Opal 540, CK-Opal520, and DAPI) was used to establish the spectral library of fluorophores required for multispectral analysis. Individual cells were identified by detecting nuclear spectral elements (DAPI). For co-expression analysis, the data obtained using InForm 2.2.1 were sent to TIBCO Spotfire, and the threshold for the positivity of each marker was determined using the IHC scoring method. For each antibody, all cells in each slide were counted (positive and negative), and the data were categorized and exported to an Excel file (Microsoft Corp., Redmond, WA, USA) for analysis. The proportions of CD8-, Foxp3-, CD68-, CD20-, CD4-, and CK-positive cells in the regions of interest (ROIs) in each slide were calculated. The total number of positive cells were counted as the total immune cell infiltrations in the tissues. The percentage of each immune cell subset was calculated by dividing the absolute number of each subset by the total number of cells.

### 2.5. Region of Interest (ROI)

The ROIs were defined as follows (Figure 1):

Cortical interstitium: Mononuclear cell-infiltrated area in the renal cortex except the area surrounding a large vessel with no severe tubular atrophic or fibrotic changes; representative area for Banff score “i”.

Area surrounding large vessel: Mononuclear cell-infiltrated area around veins, arteries, or lymphatics. These areas were not considered to be not meaningful for assessment of Banff score “i”. We did not separate each.

Medullary ray: The area with structures consisting of bundles of renal tubules which are formed in the renal cortex and continue to run through the renal medulla as medullary striations. This area was not separated from the cortical interstitium in Banff scoring. 

Medulla: Mononuclear cell-infiltrated area in the renal medulla.

Subcapsular area: The area immediately beneath the renal capsule, which may show nonspecific scarring or inflammation. The condition is thought to be related to surgical handling.

The rectangular ROIs were marked on the scanned images acquired after multiplex immunofluorescence staining (Figure 2).

Positive and negative cell counting was performed independently within these rectangular ROIs. Then, the total number in each rectangular ROI was assessed. Any rectangular ROI with less than a total of 100 cells was excluded (Figure 3). 

In acute TCMR cases (*n* = 9), the cortical interstitium (*n* = 16), area surrounding a large vessel (*n* = 13), medullary ray (*n* = 8), medulla (*n* = 10), and subcapsular area (*n* = 3) were selected for evaluation. In BKVAN cases (*n* = 5), the cortical interstitium (*n* = 12), area surrounding a large vessel (*n* = 3), medullary ray (*n* = 2), medulla (*n* = 3), and subcapsular area (*n* = 3) were selected for evaluation.

### 2.6. Statistical Analysis

The analyses were performed using R version 4.1.2. (The R Foundation for Statistical Computing; Auckland, New Zealand). The difference between mononuclear cell subsets in relation to different regions of interest (ROIs) for acute TCMR and BKVAN cases was calculated using the Kruskal–Wallis rank sum test. The difference of mononuclear cell subsets between acute TCMR and BKVAN was calculated using the Wilcoxon rank sum test. Kendall’s rank correlation test was performed to evaluate the association between graft age and mononuclear cell proportion in acute TCMR cases. Statistical analysis could not be performed for the subscapular area because of the small number of samples.

## 3. Results

The proportions of mononuclear cell subsets in different regions of interest (ROIs) are presented and compared in terms of various aspects of acute TCMR and BKVAN infiltrations in Table 3.

### 3.1. The Proportions of Mononuclear Cell Subsets in Acute TCMR Cases in Relation to Different Regions of Interest (ROIs) 

In acute TCMR cases, the cortical interstitium (*n* = 16), area surrounding a large vessel (*n* = 13), medullary ray (*n* = 8), and medulla (*n* = 10) were selected as ROIs for the analysis. The subcapsular area (*n* = 3) was excluded from the analysis because of the small number of samples. In the cortical interstitium, the most frequently observed cells were CD68+ cells (41.8%; Table 3), followed by CD8+, CD20+, Foxp3, and CD4+. In the medullary rays, the most frequently observed cells were CD8+ (38.2%; Table 3), followed by CD68+, CD20+, Foxp3, and CD4+ cells. In the medulla, the most frequently observed cells were CD8+ (41.2%; Table 3), followed by CD68+, CD20+, Foxp3, and CD4+ cells. In the area surrounding a large vessel, the most frequently observed cells were CD20+ (32.6%; Table 3), followed by CD8+, CD68+, Foxp3, and CD4+ cells. The CD68+ cell had a significantly higher prevalence in cortical interstitium and medulla areas than the medullary ray or area surrounding a large vessel (41.8%, 36.8% vs. 21.2%, 26.9% *p* = 0.0165; Table 3). The proportion of CD20+ cells differed significantly between regions of interest (*p* = 0.0024; Table 3). The CD20+ cell had a significantly higher prevalence in the area surrounding a large vessel than the other ROIs (32.6% vs. 11.3%, 21.0%; Table 3).

### 3.2. The Proportions of Mononuclear Cell Subsets in BKVAN Cases in Relation to Different Regions of Interest (ROIs) 

In BKVAN cases (*n* = 5), the cortical interstitium (*n* = 12), area surrounding a large vessel (*n* = 3), medullary ray (*n* = 2), and medulla (*n* = 3) areas were selected as ROIs for comparison with acute TCMR cases. In the cortical interstitium, the most frequently observed cells were CD68+ cells (40.1%; Table 3), followed by CD8+ (22.8%), CD20+ (16.4%), CD4+ (3.4%), and FOXP3+ cells (1.7%). In the medullary rays, the most frequently observed cells were CD8+ (36.7%; Table 3), followed by FOXP3+ (22.4%), CD68+ (18.7%), CD4+ (15.3%), and CD20+ cells (6.8%). In the medulla, the most frequently observed cells were CD20+ (51.9%; Table 3), followed by CD68+ (28.1%), CD8+ (20.3%), FOXP3+ (2.5%), and CD4+ cells (0.9%). In the area surrounding a large vessel, the most frequently observed cells were CD68+ (26%; Table 3), followed by CD8+ (25.8%), CD8+ (18.3%), CD4+ (5.6%), and FOXP3+ cells (4.8%). The CD20+ cell had a significantly higher prevalence in the medulla (51.9%) when compared with other ROIs. The proportion of CD20+ cells in ROI also differed significantly between areas (*p* = 0.049; Table 3).

### 3.3. The Proportions of Mononuclear Cell Subsets in Acute TCMR versus BKVAN cases

In the cortical interstitium, CD8+ cells were significantly more prevalent in acute TCMR cases than BKVAN cases (34% vs. 22.8%, *p* = 0.034; Table 3 and Figure 4). In the medullary ray, there was no significant difference between acute TCMR and BKVAN cases with respect to the proportion of mononuclear subsets. (Table 3 and Figure 5). In the medulla, CD20+ cells were significantly more prevalent in BKVAN cases than acute TCMR cases (51.9% vs. 11.3%, *p* = 0.028; Table 3 and Figure 6). In the area surrounding a large vessel, there was no significant difference between acute TCMR and BKVAN cases with respect to the proportion of mononuclear subsets. (Table 3 and Figure 7).

## 4. Discussion

The multiplex immunostaining of TCMR samples showed that CD68+ cells were more commonly observed than other types of lymphocytes. Conventional IHC studies have previously suggested this, but we could prove it by using compositional analysis. The allorecognition is the first step in a sequence of complex events that lead to T-cell activation, antibody production, and rejection [12,13]. Several studies have demonstrated the predominant occurrence of monocytes/macrophages, in addition to T cells, using more advanced immunohistochemistry in TCMR cases [14,15,16,17]. Girlanda et al. [18] previously demonstrated that a large number of infiltrating monocytes was associated with renal dysfunction. High levels of monocyte infiltration in rejecting allografts have been linked to severe rejection, and glomerular monocyte infiltration has been suggested as an indicator of poor grafting outcomes [19,20]. Therefore, IHC monocyte and T-cell percentile measurements may be useful in the assessment of acute clinical findings, especially graft biopsies. Consequently, the macrophage targeting agents have been considered a rejection therapeutic option for improving outcomes for transplant recipients [21,22].

The medulla has been considered as the least specific region for rejection-induced injury. Medullary inflammation may be associated with different diseases, including pyelonephritis and interstitial nephritis [23,24]. However, Wang et al. [25] suggested that medullary inflammation should not rule out a rejection, because medullary inflammation may be a “spillover effect” of cortical lesions, and the medulla has a lower rejection sensitivity (approximately 77% of that in the cortex). Sis et al. [23] insisted that acute rejection-related lesions are more common and severe in the cortex, while the renal medulla does not sufficiently reflect the inflammation associated with cortical rejection. The positive and negative indicators of inflammatory change in the medulla during the allograft rejection are insufficient as possible predictors. We also tested the composition of infiltrating leukocyte subsets in medulla and medullary rays in acute TCMR cases. Infiltrating leukocyte subsets have well-defined anatomic structures consisting of bundles of renal tubules. The bundles are formed in the renal cortex and continue to run through the renal medulla. In the medullary rays, the most frequently observed cells were CD8+ (38.2%), followed by CD68+(21.2%), CD20+(21%), Foxp3, and CD4+ cells. In the medulla, the most frequently observed cells were CD8+ (41.2%), followed by CD68+ (36.8%), CD20+ (11.3%), Foxp3, and CD4+ cells. The prevalence of CD68+ cells was significantly different across the cortical interstitium (41.8%), medulla (36.8%), medullary ray (21.2%), and area surrounding a large vessel (26.9%) ROIs (*p* = 0.0165; Table 3 and Figure 4). Moreover, the proportions of CD20+ cells were significantly different between the medulla (11.3%) and medullary ray (21.0%) areas (*p* = 0.0024; Table 3 and Figure 4). In the medullary ray area, a significant difference was found between then proportions of mononuclear cell subsets found in the medulla and cortical interstitium. Our results demonstrate that medullary ray is an important histologic area. Because medullary rays are located in the cortex, this structure has been identified as a reliable area for inflammation-linked diagnoses and estimation of infiltration intensity using Banff scores. Our results suggest that it is necessary to assess the proportion of medullary ray inflammation on the Banff score chart, especially if the proportion is larger than usual in biopsy specimens.

BKVAN diagnosis has been derived from histological assessments of lymphocytic interstitial infiltrates and the nuclear reaction to the anti-SV-40 antibody—a marker of viral replication [2,7]. Notably, BKVAN diagnosis is difficult to differentiate from acute TCMR. Our data indicate that mononuclear cell subsets in BKVAN samples differ from those of acute TCMR. In the cortical interstitium, CD8+ cells were significantly more prevalent in acute TCMR cases compared to BKVAN cases (34% vs. 22.8%, *p* = 0.034; Table 3 and Figure 6). In the medulla, CD20+ cells were significantly more prevalent in BKVAN cases compared to acute TCMR cases (51.9% vs. 11.3%, *p* = 0.028; Table 3). However, further investigation is required to confirm the differences between the two diagnoses using the multiplex immunohistochemistry method.

According to the 1997 Banff guidelines update, assessment areas that should be excluded from Banff lesion scoring include “fibrotic areas, immediate subcapsular cortex, adventitia around large veins, and lymphatics” [1,3,6,7]. However, differences between inflammatory cell subsets in those, and in non-scarred cortical areas, were not investigated. We detected differences in cell subset composition in areas near large vessels (*n* = 16) from the cortical interstitium in acute TCMR cases. CD20+ cells (32.6%) were the most frequently observed mononuclear cells, although were not associated with acute TCMR effects. We also counted mononuclear cells in the subcapsular area (*n* = 3). Although CD20+ cells were the dominant subset in all these areas, the results were not statistically significant due to the insufficient number of samples.

In summary, this multiplex immunofluorescence assay was a useful method for analyzing mononuclear cell subsets in a renal allograft biopsy. The most frequently observed cells in the cortical interstitium of acute TCMR cases were CD68+ macrophages or monocytes rather than T cells. These cells may contribute to allograft damage in the course of rejection. Although it was difficult to distinguish between BKVAN and acute TCMR by using multiplex immunofluorescence assays, the ratio of CD8+ cells was higher in acute TCMR compared to of BKVAN cases in cortical interstitium (34% vs. 22.8%, *p* = 0.034). In the medulla, the assay showed a significantly higher proportions of CD20+ cells in BKVAN cases compared to acute TCMR cases (51.9% vs. 11.3%, *p* = 0.028) However, the cellular proportions may be influenced by many factors, e.g., differences in graft age, patient age, and individual differences in immune regimens or doses during the whole clinical course. Further studies that observe change over time in the same patient after applying this method to the protocol biopsy will be important. In areas surrounding large vessels, previously thought of as non-diagnostic, the CD20+ cells were dominant. In the medullary rays, cellular subsets were similar to those of the medulla despite being located in the cortex. Therefore, if cellular infiltration in the medullary rays is high or the medullary ray area occupies a lot of the cortical area, Banff “i” scoring should be carried out more carefully given our finding that cellular proportions in the medullary ray were different from those in the renal cortical interstitium.

## Figures and Tables

**Figure 1 diagnostics-12-00268-f001:**
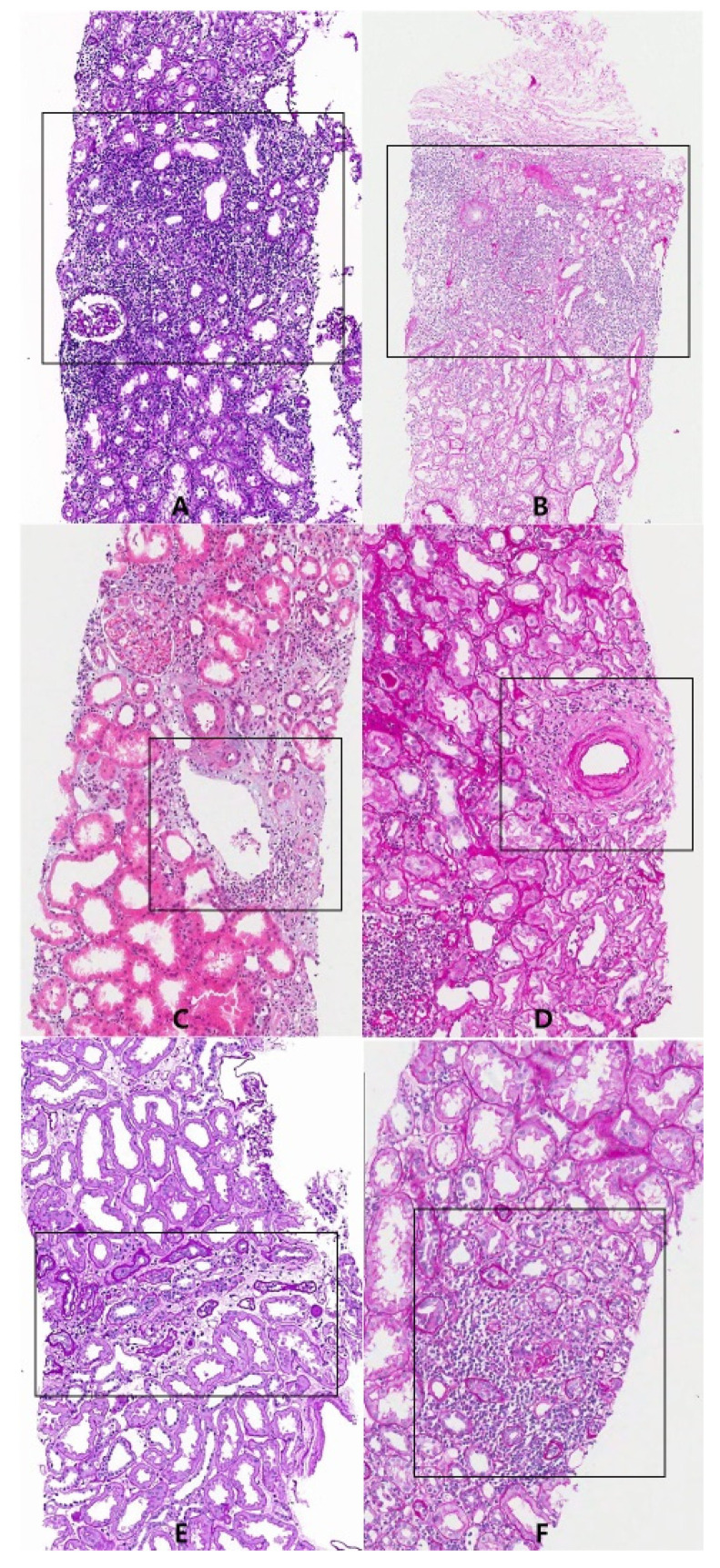
Representative areas of regions of interest (ROIs). (**A**) Cortical interstitial infiltration area (periodic acid–schiff stain, ×100); (**B**) subcapsular area (periodic acid–schiff stain, ×100); (**C**) area surrounding the vein (hematoxylin and eosin stain, ×100); (**D**) adventitia area of artery (periodic acid–schiff stain, ×100); (**E**) medullary ray, longitudinal section (periodic acid–schiff stain, ×100); (**F**) medullary ray, cross section (periodic acid–schiff stain, ×100).

**Figure 2 diagnostics-12-00268-f002:**
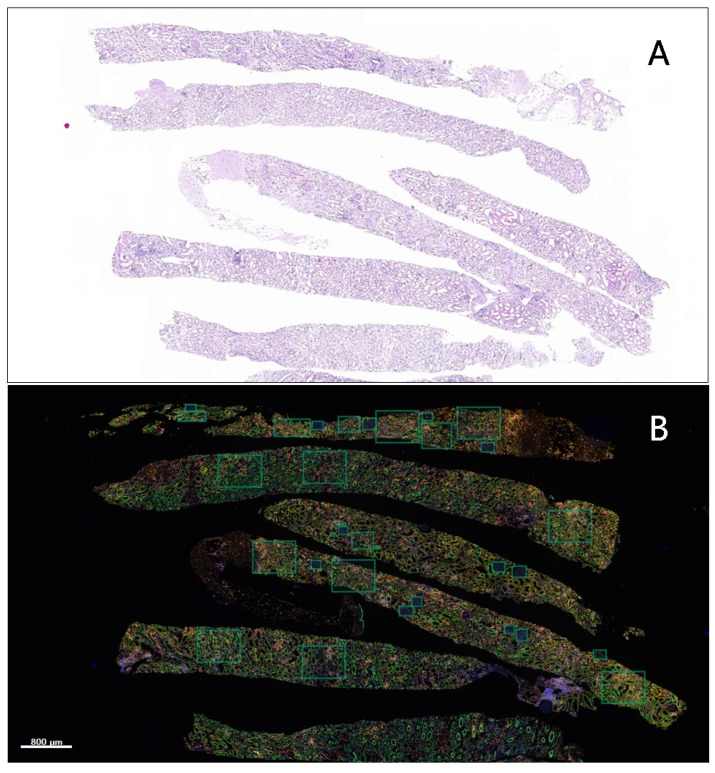
Selection of regions of interest (ROIs). (**A**) Acute T-cell-mediated rejection samples were stained with PAS; (**B**) the scanned images show multiplex immunofluorescence staining. ROIs are marked with rectangles. Cell counting was independently performed within these ROIs. The total number of cells was assessed per ROI.

**Figure 3 diagnostics-12-00268-f003:**
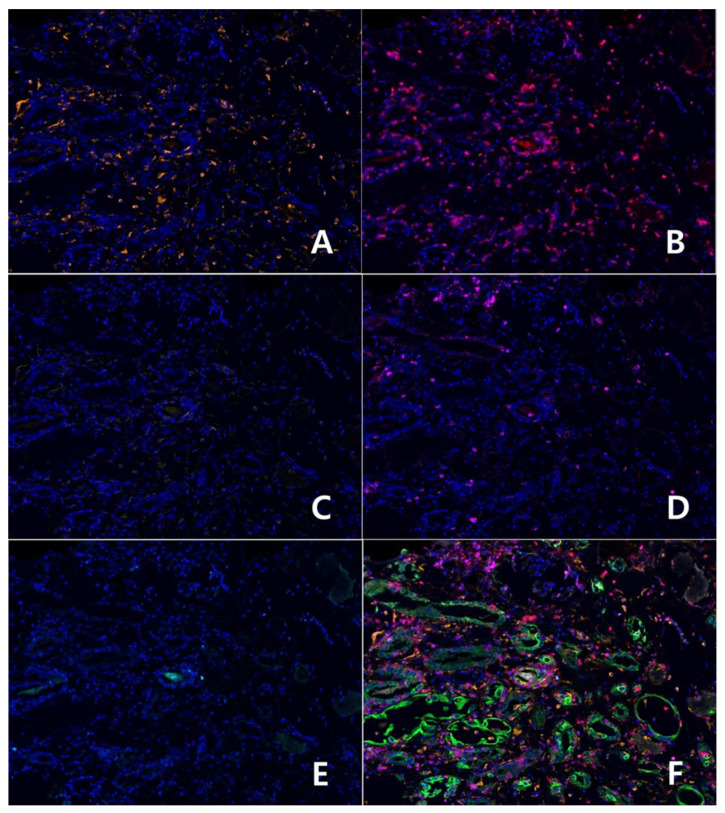
Multiplex immunofluorescent images show the individual antibody staining as follows (×200): (**A**) CD68+(Opal 620); (**B**) CD8+(Opal 690); (**C**) CD20+(Opal 570); (**D**) CD4+(Opal 540); (**E**) Foxp3+(Opal 650); (**F**) Antibody staining was merged with cytokeratin (CK, Opal 520) staining. Blue nuclear stain represents 4′,6-diamidino-2-phenylindole (DAPI).

**Figure 4 diagnostics-12-00268-f004:**
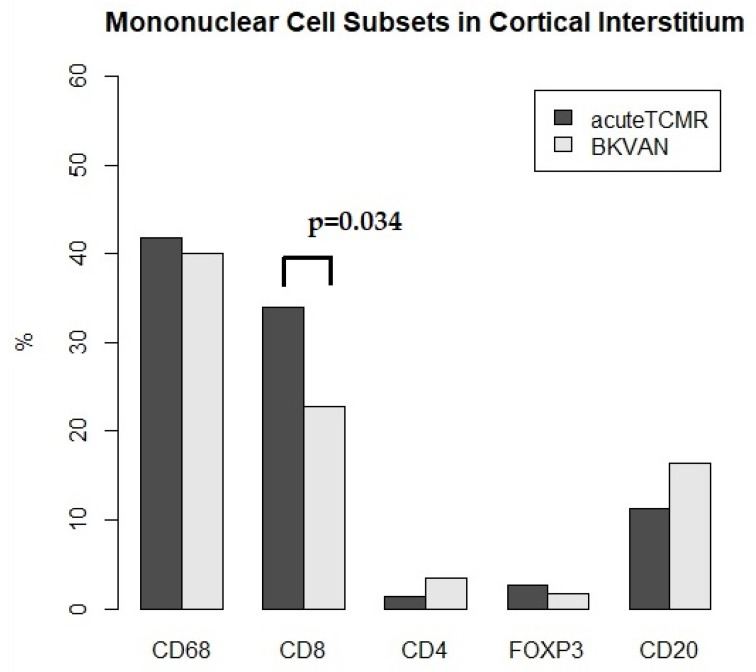
The proportions of mononuclear cell subsets for acute TCMR versus BKVAN cases in cortical interstitium. CD8+ cells were significantly more prevalent in TCMR than BKVAN cases (*p* = 0.034).

**Figure 5 diagnostics-12-00268-f005:**
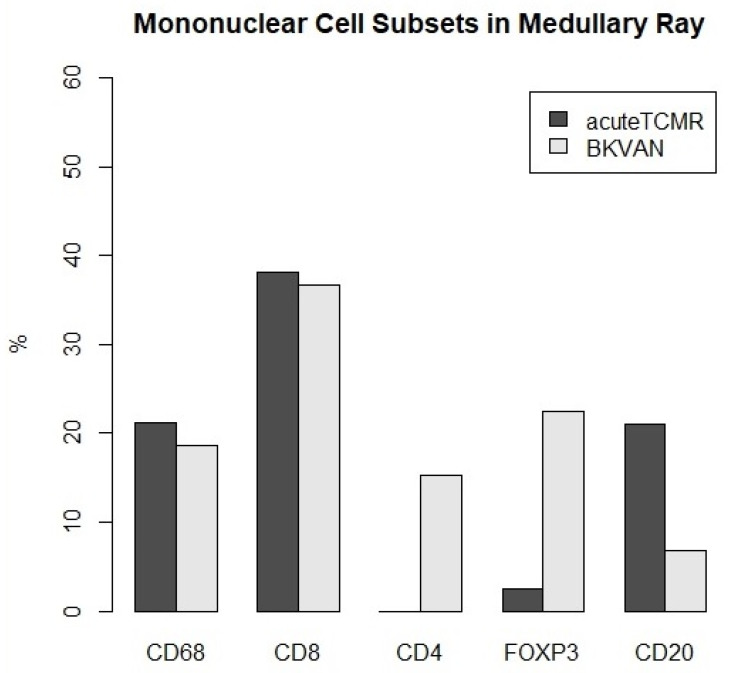
The proportions of mononuclear cell subsets for acute TCMR versus BKVAN cases in medullary ray.

**Figure 6 diagnostics-12-00268-f006:**
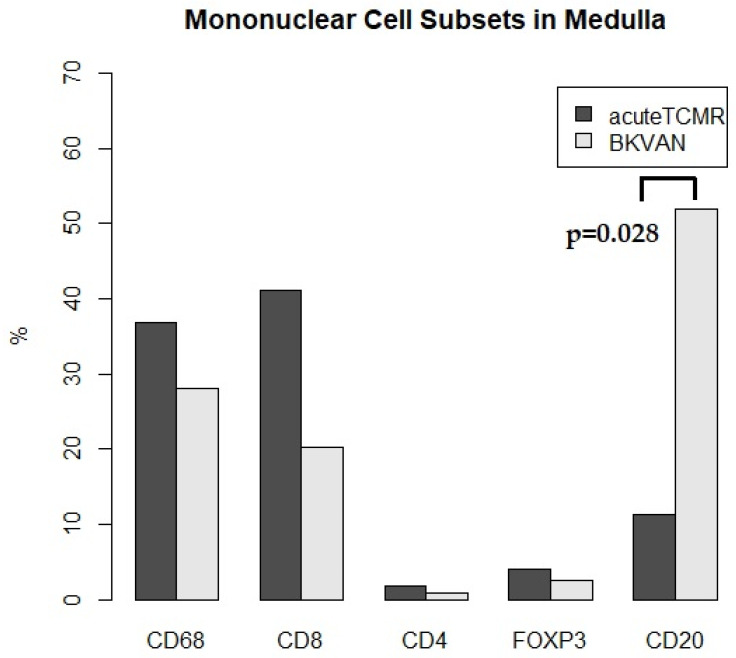
The proportions of mononuclear cell subsets for acute TCMR versus BKVAN cases in medulla. CD20+ cells were significantly more prevalent in BKVAN than TCMR cases (*p* = 0.028).

**Figure 7 diagnostics-12-00268-f007:**
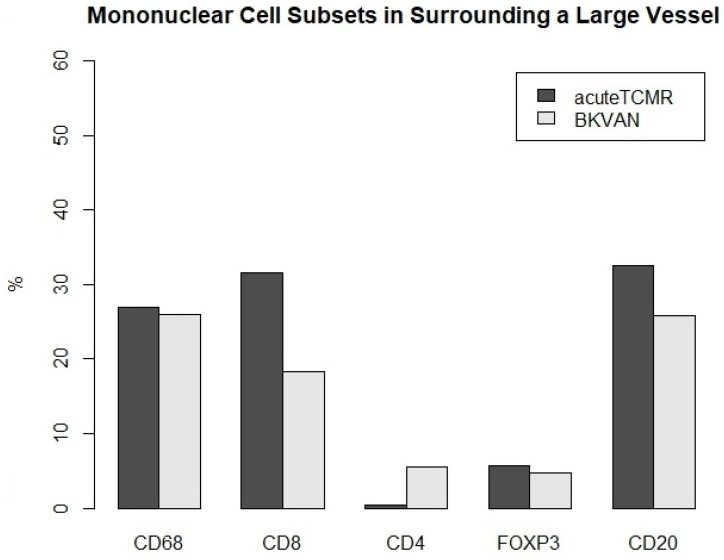
The proportions of mononuclear cell subsets in acute TCMR versus BKVAN cases in the surrounding a large vessel ROI.

**Table 1 diagnostics-12-00268-t001:** Basic data of acute TCMR cases.

Case	Age	Sex	Graft Age	Creatinine	Banff Score	Grade
			Months	mg/dL		
1	42	F	3	1.8	i2, t3, ptc1, ti2	1A
2	33	M	15	4.3	i2, t3, ptc1, ci2, ct2	1B
3	40	M	9	4.4	i3, t3, v2	2B
4	33	M	8	2.7	i2, t2, ptc1	1A
5	55	M	1	1.2	i2, t2, v2, ptc1	2B
6	56	F	10	2.4	i2, t3, cv1	1B
7	58	M	4	2.8	i3, t2, ptc1, ti3	1A
8	62	M	8	2.4	t1, v2, ptc1, ci1, ct1, ti2, i-IFTA3	2B
9	42	F	7	4.5	i3, t3, g1, ci2, ct2, cv1, ah1, ti3	1B

**Table 2 diagnostics-12-00268-t002:** Basic data of BK virus nephropathy cases.

Case	Age	Sex	Graft Age	Creatinine	Banff Score	Stage
			Months	mg/dL		
1	62	M	5.5	2.2	ci1, ct1, cv1, i-IFTA1	B
2	58	M	2.5	1.8	i3, t1, ptc1, ti3	B
3	30	M	7	1.8	ci1, ct1, ti3	A
4	25	M	4	3.3	I2,ci1,ct1,ti3	B
5	62	M	6.5	3.1	i2, ptc1, ci2, ct3, ti3, i-IFTA3	C

**Table 3 diagnostics-12-00268-t003:** Median mononuclear cell subset numbers and ratios counted in regions of interest (ROI) and assessed using five monoclonal antibodies in nine cases of acute T-cell-mediated rejection (Acute TCMR) and five cases of BK virus-associated nephropathy (BKVAN).

	Acute TCMR	Acute TCMR	Acute TCMR	Acute TCMR	BKVAN	BKVAN	BKVAN	BKVAN
	Cortical Interstitium	Medullary Ray	Medulla	Area Surrounding Large Vessel	Cortical Interstitium	Medullary Ray	Medulla	Area Surrounding Large Vessel
	(*n* = 16)	(*n* = 8)	(*n* = 10)	(*n* = 13)	(*n* = 12)	(*n* = 2)	(*n* = 3)	(*n* = 3)
Count	Median (IQR)	Median (IQR)	Median (IQR)	Median (IQR)	Median (IQR)	Median (IQR)	Median (IQR)	Median (IQR)
CD68	345.5	284	365.5	283	552	275	913	335
	(111.0–868.8)	(58.5–404.5)	(125.5–1242.0)	(102.0–501.0)	(82.0–955.0)	(151.5–398.5)	(776.5–1538.0)	(223.5–695.5)
CD8	316.5	252	410	322	321.5	486	552	1522
	(80.5–600.3)	(69.5–496.5)	(195.0–1919.0)	(132.0–691.5)	(42.5–1041.0)	(329–643)	(468.5–1990.5)	(762–1604.5)
CD4	8	0	43	2	61	126	24	226
	(1.0–30.8)	(0–160.3)	(5.0–147.3)	(0–51.5)	(2.3–697.5)	(69.5–182.5)	(12–240)	(113.5–504.5)
FOXP3	39.5	32.5	27.5	49	21	190.5	68	197
	(4.8–69.8)	(11.0–55.3)	(7.5–285.8)	(19.5–109.5)	(3.5–623.3)	(117.75–263.25)	(43–175)	(99–1860)
CD20	137	370	209.5	210	329.5	93	1423	900
	(27.5–476.0)	(11.5–488.0)	(26.8–423.8)	(69.5–872.0)	(39.5–689.8)	(59.5–126.5)	(1391.5–1430.5)	(501–1524)
Total	902.5	1172	1127	1111	1753	1170.5	2739	4066
	(276.8–2117.5)	(179.0–1906.3)	(304.3–4583.8)	(374.5–2171.0)	(205.8–4286.5)	(985.75–1355.25)	(2730.5–5214.5)	(2142–6188.5)
Ratio	Median (IQR)	Median (IQR)	Median (IQR)	Median (IQR)	Median (IQR)	Median (IQR)	Median (IQR)	Median (IQR)
CD68 (%)	41.8	21.2	36.8	26.9	40.1	18.7	28.1	26
	(31.7–46.9)	(19.3–30.9)	(21.0–42.8)	(18.1–34.2)	(23.4–48.7)	(11.1–26.3)	(25.8–30.7)	(15–38.7)
CD8 (%)	34	38.2	41.2	31.6	22.8	36.7	20.3	18.3
	(24.7–45.1)	(18.3–41.8)	(30.3–46.2)	(18.5–39.2)	(13.6–30.3)	(29.1–44.3)	(17.2–32.4)	(9.6–29.9)
CD4 (%)	1.3	0	1.7	0.4	3.4	15.3	0.9	5.6
	(0.2–6.6)	(0–14.5)	(0.7–21.7)	(0–1.8)	(0.7–32.3)	(8.1–22.6)	(0.4–3.4)	(3–7.5)
FOXP3 (%)	2.7	2.5	4	5.7	1.7	22.4	2.5	4.8
	(1.5–6.8)	(1.3–20.1)	(1.1–6.1)	(3.4–15.2)	(1.2–15.7)	(12.7–32.2)	(1.6–3.1)	(2.7–23.6)
CD20 (%)	11.3	21	11.3	32.6	16.4	6.8	51.9	25.8
	(3.7–23.1)	(8.5–34.3)	(5.2–16.8)	(18.8–41.8)	(7.2–25.3)	(5–8.6)	(34.8–52.4)	(24–36.3)

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
