# Peer review of "Multiplex Immunofluorescence Assay of Infiltrating Mononu-Clear Cell Subsets in Acute T-Cell-Mediated Rejection and BK Virus-Associated Nephropathy in the Allograft Kidney"

_diagnostics, 2022, doi:10.3390/diagnostics12020268_

Round 1

Reviewer 1 Report

The authors presented results by using an innovative technology of multiplex immunofluorescence staining to identify infiltrating mononuclear cell subsets  in rejected transplanted kidney sections in comparison with BK virus-associated nephropathy. The authors nicely presented that the multiplex immunofluorescence staining is a reliable technic to assess immunologic events in this organ. The analysis, in accordance with findings in referenced papers, shows monocyte lineage infiltration in the kidneys of these patients.

My comments are:

(1) The authors mention that the tissue samples are from various time points after transplantation as it is also shown in Table 1 and 2. They also discuss in the Discussion section that " this multiplex immunofluorescence assay would be useful to follow up on the immunologic environmental changes in the allograft because these cellular proportions may be changed during the course of rejection or treatment. Based on the median value, there seems to be a significant difference in the cell count/ratio of the same cell type among the patients. Can the authors suggest what the reason for that, e.g. is it due to the sampling at different time point or just the individuality among the samples? Are there samples available taken at different time points from the same patients to answer this question?

(2) There is a type error in the line 187, Results section 3.3. Table 1 should be Table 3.

Reviewer 2 Report

This paper reported a utility of multiplex immunofluorescence assay of infiltrating mononuclear cell subsets, a novel method for histological evaluation, for the diagnosis of acute T-cell-mediated rejection (TCMR) and BK virus-associated nephropathy (BKVAN) in kidney allograft biopsy. The methods and the results are interesting, however, several concerns exist, especially in the lack of some data and the interpretations of the results.

The reviewer considers that the major missions of this study are follows:

  1. To characterize the difference in the infiltrating mononuclear cell subsets between in TCMR and BKVAN
  2. To characterize the difference in the infiltrating mononuclear cell subsets between in the cortex and medulla in TCMR
  3. To characterize the significance of medullary ray (MR) in TCMR and BKVAN

For the first mission, the authors should compare the infiltrating mononuclear cell subsets in total cortex, MR and medulla, respectively, and figure out the significant differences.

For the second mission, the authors should compare the infiltrating mononuclear cell subsets between in total cortex and medulla, and between in MR and medulla.

For the third mission, the authors should compare the infiltrating mononuclear cell subsets between in total cortex and MR in TCMR and BKVAN, independently.

In addition, the reviewer cannot understand the exact meaning of the comparison between in the cortical interstitium versus area surrounding large vessels in TCMR, which was performed by the authors and described in the Results 3.4 in the manuscript. The reviewer considers that the comparison between the cortical interstitium and the area surrounding large vein, not artery, might be interesting because the significance of the perivenular mononuclear cell infiltration sometimes observed in the allograft is still controversial for the differentiation from rejection. The infiltrates surrounding the large artery usually do not have any senses in the rejection, except for the area of vascular rejection itself.

Considering the situations mentioned above, the reviewer requests the authors to consider the following points and revise the manuscript.

<Major points>

  1. The data in medullary ray and medulla in BKVAN should be demonstrated in Table 3. The required comparisons should be performed and described in the results, figures (if they are required) and discussion.
  2. The data in the area surrounding the large vein in TCMR should be demonstrated in Table 3, instead of the data in the area surrounding the large vessels. And the required comparisons should be performed and described in the results and discussion.
  3. In the abstract, the authors concluded that the cellular subsets were similar in acute T-cell-mediated rejection and BK virus-associated nephropathy, however, the reviewer thinks there are some significant difference as the authors demonstrated the higher levels of CD8+ cells in TCMR and higher levels of CD20+ cells in BKVAN.
  4. In the Figure 4 and Figure 5, some of the bars are not reflecting the values described in the Table 3. Please draw the figures exactly.
  5. In the discussion (line 231-233), the authors described that “The proportion of mononuclear cell subsets in the medullary rays did not differ from that in the outer medulla. Therefore, inflammatory markers in the medullary ray or medulla cannot serve as reliable indicators of acute rejection changes.”, however, the reviewer considered that some differences exist in the proportions of CD68+ and CD20+ cells, although the reviewer is not confident that the comparison between MR and medulla is important or not in order to demonstrate the anatomical significance of MR in the cortex.
  6. In the discussion (line 244-245), the authors described that “Our data indicate that mononuclear cell subsets in BKVAN samples did not differ from those of acute TCMR.”, but the reviewer considers that some differences exist in the proportions of CD8+ and CD20+ cells, as mentioned in the comment 3.
  7. In the summary of the discussion (line 266-268), the authors described that “BKVAN could not differentiate from acute TCMR by this multiplex immunofluorescence assay, because both had similar cellular infiltration subsets.”, but it is not true because of the same reason in the comment 3 and comment 6.
  8. In the summary of the discussion (line 274-277), the authors described that “Therefore, the proportion of medullary rays in the cortical interstitium should be considered for Banff “i” scoring because the cellular proportions were different from those of the renal cortical interstitium and the same as those of the medullary area.” This conclusion is misleading and I cannot understand the logic of this conclusion. The conclusion on the significance of MR should be revised after the careful comparison of the cellular subset in MR versus the total cortex in the medulla in TCMR.

<Minor points>

  1. In line 18, delete hyphen of “cy-tokeratin”.
  2. In line 107, delete hyphen of “re-quired”.
  3. In line 169, delete “2.1 Cortical in” before “3.1 Cortical……”.

Round 2

Reviewer 2 Report

The authors sincerely respond to my comments and they presented additional data and comparisons clearly. I appreciate their efforts, however, the quality of the figures presented in the revised manuscript got worse and most of them show no significant meaning.

<Major points>

  1. So, I would like to request the authors that some of the figures (Fig. 4 and Fig. 5) should be deleted and the essential figures (like Fig.6, 7, 8, 9) should be changed to the former style of the bar chart (like the Fig. 5 in the original manuscript) . That will improve the manuscript remarkably.
  2. In addition, the results about the comparisons of cell subsets in relation to graft age (Results 3.4., Fig. 10, and Fig 11) are not necessary in this study, so I recommend them to be deleted.

<Minor point>

  1. P4 , line 135: The area was not considered to be not meaningful for assessment of Banff score “i.” The term “not” appeared twice, is it correct?

Author Response

The authors sincerely respond to my comments and they presented additional data and comparisons clearly. I appreciate their efforts, however, the quality of the figures presented in the revised manuscript got worse and most of them show no significant meaning.

<Major points>

So, I would like to request the authors that some of the figures (Fig. 4 and Fig. 5) should be deleted and the essential figures (like Fig.6, 7, 8, 9) should be changed to the former style of the bar chart (like the Fig. 5 in the original manuscript) . That will improve the manuscript remarkably.
In addition, the results about the comparisons of cell subsets in relation to graft age (Results 3.4., Fig. 10, and Fig 11) are not necessary in this study, so I recommend them to be deleted.

I removed figure 4 and 5. I changed figure 6,7,8,9 to former style. I removed result 3.4 and figure 10 and figure 11.

<Minor point>

P4 , line 135: The area was not considered to be not meaningful for assessment of Banff score “i.” The term “not” appeared twice, is it correct?

I removed first "not", thank you for your kind response ->The area was considered to be not meaningful for assessment of Banff score “i."

I sincerely appreciated for your advises.